# Epigenetic Mechanisms of Epidermal Differentiation

**DOI:** 10.3390/ijms23094874

**Published:** 2022-04-28

**Authors:** Chiara Moltrasio, Maurizio Romagnuolo, Angelo Valerio Marzano

**Affiliations:** 1Dermatology Unit, Fondazione IRCCS Ca’ Granda Ospedale Maggiore Policlinico of Milan, 20122 Milan, Italy; maurizio.romagnuolo@unimi.it (M.R.); angelo.marzano@unimi.it (A.V.M.); 2Department of Medical Surgical and Health Sciences, University of Trieste, 34137 Trieste, Italy; 3Department of Pathophysiology and Transplantation, Università degli Studi di Milano, 20122 Milan, Italy

**Keywords:** keratinocytes, epidermal differentiation complex, epigenetic regulators, ATP-dependent chromatin remodeler, DNA methyltransferases, histone modifications, polycomb proteins, microRNAs, psoriasis, atopic dermatitis

## Abstract

Keratinocyte differentiation is an essential process for epidermal stratification and stratum corneum formation. Keratinocytes proliferate in the basal layer of the epidermis and start their differentiation by changing their functional or phenotypical type; this process is regulated via induction or repression of epidermal differentiation complex (EDC) genes that play a pivotal role in epidermal development. Epidermal development and the keratinocyte differentiation program are orchestrated by several transcription factors, signaling pathways, and epigenetic regulators. The latter exhibits both activating and repressive effects on chromatin in keratinocytes via the ATP-dependent chromatin remodelers, histone demethylases, and genome organizers that promote terminal keratinocyte differentiation, and the DNA methyltransferases, histone deacetylases, and Polycomb components that stimulate proliferation of progenitor cells and inhibit premature activation of terminal differentiation-associated genes. In addition, microRNAs are involved in different processes between proliferation and differentiation during the program of epidermal development. Here, we bring together current knowledge of the mechanisms controlling gene expression during keratinocyte differentiation. An awareness of epigenetic mechanisms and their alterations in health and disease will help to bridge the gap between our current knowledge and potential applications for epigenetic regulators in clinical practice to pave the way for promising target therapies.

## 1. Introduction

The epidermis constitutes the skin surface and is composed of specialized epithelial cells called keratinocytes, which originate from two pools of quiescent epidermal stem cells (qESC), one living in the basal layer of the interfollicular epidermis (IFE) and the other, bulge stem cells, located in the hair follicle of the sebaceous gland that gives rise to keratinocytes of the hair follicle (HF) lineage. From qESC, transient amplifying (TA) cells generate “mature” keratinocytes residing in the basal layer of the epidermis or in the hair bulge (region of the outer root sheath) [1]. Progenitor cells of the basal layer undergo a programed process of proliferation and differentiation, culminating in the formation of a cornified envelope consisting of corneocytes, and terminally differentiated, enucleated, keratinocytes [2]. Epidermal growth and differentiation are driven by the accurate expression of multiple genes regulated by several transcriptional and post-transcriptional mechanisms [3]. This strict control is often challenged by external biological or environmental factors, such as UV light [4], and/or modulated by physiological factors (e.g., aging) [5]. Collectively, genomic alterations result in a broad deregulation of gene expression and the dysfunction of signal transduction pathways that control proliferation and several cellular functions. In addition to changes in DNA sequence and structure, it has become increasingly evident that all cell processes can be deeply orchestrated by epigenetic mechanisms [6]. Epigenetic modifications refer to reversible changes in gene expression without alteration of the DNA sequence and occur throughout all stages of development or in response to environmental factors [6]. On the other hand, recent research has raised the notion that epigenetic mechanisms could mediate “stable” changes in tissue function. These studies converged on a set of common enzymatic modifications to chromatin structure that can up- or down-regulate gene expression. Thus, epigenetic mechanisms play a crucial role in the control of cellular functions and can help to explain the relationships between a genetic background and effects of the environment on the susceptibility to different diseases [7], including skin disorders [8]. During last years, it has been shown that epigenetic mechanisms are involved in the growth and differentiation of keratinocytes by regulating a gene-rich region at 1q21 known as the epidermal differentiation complex (EDC) [9]. These genes encode involucrin, loricrin and small proline-rich proteins, as well as several calcium binding proteins and S100A proteins [10] that play a pivotal role in epithelial tissue development and repair by regulating the terminal differentiation program of keratinocytes through signal transduction events [9]. During terminal differentiation, a pool of epigenetic regulators influences the programing of epidermal keratinocytes via induction or repression of EDC genes, thus playing a crucial role in epidermal development. In particular, the program of epidermal development and keratinocyte differentiation is governed by several transcription factors (the most important of which is p63), signaling pathways (Notch, Wnt, Bmp, Hedgehog, and so on), and epigenetic regulators. The latter exhibits both activating and repressive effects on chromatin in keratinocytes and includes the following: ATP-dependent chromatin remodeler Brg1 and Mi-2β; histone demethylase Jmjd3 and Setd8 and genome organizer Satb1 that promote terminal keratinocyte differentiation; DNA methyltransferase DNMT1; histone deacetylases HDAC1/2; Polycomb proteins (in particular, Bmi1 and Ezh1/2) that stimulate the proliferation of progenitor cells and inhibit the premature activation of terminal differentiation-associated genes [11,12,13,14,15]. Despite the advancement of knowledge, many aspects of epigenetic control of the gene expression programs in keratinocytes remain to be elucidated.

In the present review, we aim to summarize the available evidence on activating and repressive effects on chromatin that promote epidermal differentiation. In the first section, we review the normal functions of epigenetic mechanisms and their role in epidermal differentiation. The second section focuses on our current understanding of the different types of epigenetic changes in psoriasis and atopic dermatitis—as emblematic examples of inflammatory skin diseases—providing some specific examples of each.

In summary, we provide an overview of the epigenetic mechanisms underlying epidermal differentiation in health and disease and we conclude with a discussion of the potential implications for clinical practice.

## 2. Search Strategy 

### Criteria for Paper Selection

To provide an overview of the current state of knowledge regarding the topic of epidermal differentiation, papers were selected from those included in the electronic databases PubMed/MEDLINE, Google Scholar, Scopus, and Web of Science over the past 20 years. The following search terms were used: keratinocytes, epigenetics, epigenetic regulators, methylation, histones, epidermal differentiation, epidermal differentiation complex, microRNAs, psoriasis, atopic dermatitis, inflammatory skin diseases, environmental factors. Each selected paper was analyzed, the data extracted, and presented in the main text to provide an overview of the main aspects of epigenetic regulation that influences the programming of epidermal keratinocytes.

## 3. The Role of Epigenetics in Health

Epigenetics refers to changes in gene expression without alteration of the DNA sequence. These changes, which make up the so called “epigenome”, include modifications to DNA and the histone components of nucleosomes as well as expression of noncoding RNAs (ncRNAs). Since epigenetic signatures help determine if genes are turned on or off, it influences the production of encoded proteins, thus ensuring cell-type-specific expression [16]. Epigenetic mechanisms of the control of gene expression include several levels of regulation that are based, among others, on: (i) methylation and hydroxymethylation; (ii) histone covalent modifications, such as the addition or removal of methyl- or acetyl- groups, at specific genome regions; (iii) changes in nucleosome positioning and ATP-dependent chromatin remodeling; (iv) RNA regulation [17]. The three main epigenetic mechanisms are DNA methylation, histone modification, and miRNA. These mechanisms are responsible for the initiation and maintenance of a gene expression profile within a series of cellular processes, including cell differentiation, embryogenesis, and genomic imprinting [18]. DNA methylation involves the transfer of a methyl group into the C5 position of cytosine to form 5-methylcytosine and promote gene expression regulation by either recruiting proteins involved in gene repression or inhibiting the binding of transcription factor(s) to DNA [19]. During development stages, the pattern of DNA methylation changes, resulting in a plastic and dynamic landscape involving both de novo DNA methylation and demethylation. DNA methylation is regulated by a family of DNA methyltransferase (DNMTs): DNMT1, DNMT2, DNMT3A, DNMT3B, and DNMT3L [20]. DNMT1 usually methylates CpG islands on hemimethylated DNA and is localized in replication foci during the S phase of the cell cycle. In line with this function, DNMT1 is responsible for maintaining DNA methylation but can catalyze de novo DNA methylation in specific genomic contexts [21] (Figure 1a). 

Histone modification is another key epigenetic mechanism. Histone complexes are composed of two H2A-H2B dimers and a H3-H4 tetramer to form the nucleosome, and they may be chemically modified through the action of enzymes to regulate gene transcription. The most common modifications are the methylation of arginine or lysine residues, or the acetylation of lysine [22] (Figure 1b). 

Methylation can modulate the interaction between transcription factors and other proteins with nucleosomes [23], while lysine acetylation eliminates a positive charge on lysine, thereby weakening the electrostatic attraction between histone and DNA, thus resulting in partial unwinding of the DNA to make it more accessible for gene expression [22]. Mature microRNA (miRNA) is another key player in the epigenetic picture (Figure 1c). 

MiRNAs are a group of short (~22 nucleotides), single stranded, non-coding RNAs that can regulate the expression of other protein-coding gene at the post-transcriptional level. MiRNAs function via base-pairing with complementary sequences within messenger RNA (mRNA); after this binding, mRNA molecules are silenced by one or more of the following processes: (i) mRNA cleavage, (ii) destabilization of the mRNA through de-adenylation of its poly(A) tail, and (iii) translational inactivation [24]. In rare cases, miRNAs can mediate the activation of gene expression profiles under specific conditions [25].

Below, we review the mechanisms that control chromatin remodeling, with special emphasis on keratinocyte-specific gene expression programs in skin development (data illustrating the role of chromatin remodeling factors in epidermal differentiation are shown in Table 1) and in inflammatory skin diseases, such as psoriasis and atopic dermatitis. 

## 4. The Role of Epigenetic Mechanisms Involved in Epidermal Differentiation

### 4.1. Epigenetic Regulation by ATP-Dependent Chromatin Remodeling

ATP-dependent chromatin remodeling complexes use the energy of ATP hydrolysis to alter chromatin architecture by repositioning, assembling, and restructuring nucleosomes. These complexes fall into one of four families: switch/sucrose non-fermentable (SWI/SNF), Mi-2/nucleosome remodeling and deacetylase (NuRD) complex, imitation switch (ISWI)/SNF2L, and INO80 [40]. The SWI/SNF family, also called the BAF complex (Brg/Brm associated factor), is thought to regulate gene expression by altering nucleosome positioning and structure. The human SWI/SNF complex contains either Brg1 or hBrm (Brahma) as ATPase subunits that can play different roles in various cellular processes including proliferation and differentiation; moreover, these molecules also contain bromodomains that allow binding to acetylated-lysine residues within histone H3 and H4 tails [41]. Genetics studies have demonstrated a crucial role of Brg1 in the development of skin; indeed, Brg1 deficiency gravely impairs the final stage of keratinocyte terminal differentiation, leading to skin barrier defects [27]. Moreover, Brg1 promotes the relocation of the EDC locus from the nuclear periphery toward the nuclear interior and into the compartment enriched by nuclear speckles, which is associated with overexpression of the EDC genes [26]. 

Additionally, regulatory subunits of the SWI/SNF complex such as the actin-like 6A protein/Brahma-associated factor (ACTL6a/BAF53A), along with the catalytic subunits (Brg1, Brm and BAF250a), are involved in suppressing differentiation and maintaining the progenitor state in epidermal cells [28]. 

ATP-dependent chromatin remodelers of the CHD group are characterized by a chromodomain that specifically binds to methylated lysine residues; the ATPase subunits within this family include Chd1-9, but Chd3 and Chd4 (also known as Mi-2β) have a prominent role in NuRD complex. The chromatin remodeler Mi-2β is crucial for the self-renewal of epidermal precursors during the earlier phases of embryogenesis, and its ablation leads to defective basal layer formation, whereas its loss during the later stages of embryogenesis leads to impaired induction and development of the hair follicles [29]. 

### 4.2. Epigenetic Regulation by Histone Methylation

Histone methylation is the modification of certain amino acids in a histone protein by the addition of one, two, or three methyl groups. The site-specific methylation and demethylation of histone residues are catalyzed by methyltransferases and demethylases, respectively. Histone methylation is generally associated with transcriptional repression; however, methylation of some lysine and arginine residues of histones results in transcriptional activation [42]. Trimethylation of histone H3 at the lysine 4 position (H3K4me3) leads to actively transcribed genes as well as H3K4Me2/1, H3K79me2/3 and H3K36me3 [43]. 

In contrast, the chromatin of an inactive gene is enriched in H3K9me2/3, H3K27me2/3 and H4K20me3 modifications [43]. The Jumonji domain-containing protein-3 (JMJD3), also known as lysine-specific demethylase 6B (KDM6B), is a histone demethylase that regulates the trimethylation of histone H3 on lysine 27 (H3K27me3), and it has been studied extensively in immune diseases, cancer, and tumor development [44]. Recent studies illustrated that JMJD3 plays a pivotal role in the cell fate determination of pluripotent and multipotent stem cells (MSCs), enhancing their self-renewal ability and reducing the differentiation capacity of embryonic stem cells (ESCs) and MSCs into specialized cells [44]. The role of H3K27me3 in human epidermal progenitor cells has been revealed by inactivation of JMJD3, which led to a blockade of progenitor cell differentiation [30]. The genes normally induced during epidermal differentiation were repressed in epidermal progenitor cells lacking JMJD3 protein, and their induction was associated with H3K27me3 de-methylation in their promoter regions, indicating that epigenetic transcriptional upregulation by JMJD3 controls mammalian epidermal differentiation [30]. 

Histone H4 mono-methylation at lysine 20 is regulated by Setd8 histone methyltransferase. Setd8 is a transcriptional target of c-Myc, a protein involved in cellular epidermal differentiation, and it is involved in many physiological processes, including cell cycle, chromatin condensation, apoptosis, tumorigenesis, and epithelial to mesenchymal transition [45]. The loss of Setd8 methyltransferase led to inhibition of progenitor cell proliferation in the basal layer of the epidermis with simultaneous impairment of epidermal cell proliferation and differentiation [31]. Moreover, Setd8 acts as regulator of p63 expression, a master regulator of epidermal development [46], and when Setd8 is lost, epidermal cells fail to express p63 and exhibit impaired terminal differentiation, resulting in skin apoptosis [32]. 

In HaCaT cells, spontaneously immortalized human keratinocytes, knockout of *Suv39h1*, a gene encoding a H3K9 histone methyltransferase, provoked a marked reduction in the level of tri-methylation at the ninth lysine residue of the histone H3 protein and promoted changes in the expression of several keratinocyte-specific genes [33]. *Suv39h1* knockout also led to induction of genes encoding differentiation markers such as keratin 10, desmoglein 1, S100A8, and late cornified envelope (LCE1) proteins, whereas expression of genes associated with undifferentiated keratinocytes, such as keratin 14 and S100A6, remained unaltered, or at most, slightly decreased [33]. Overall, the overexpression concerned mainly genes of the middle/late differentiation stage reinforcing the keratinocyte differentiation program. 

Lysine demethylase Jarid1 family members (Jarid1a, Jarid1b, Jarid1c, and Jarid1d) demethylase the fourth lysine residue of histone H3 and participates in multiple repressive transcriptional complexes, although the specific regulatory effects on chromatin remain undefined [47]. The role of Jarid1b in cell differentiation has recently received increased attention; it positively controls epidermal cell differentiation both in vitro and in vivo, increasing expression of mesenchymal-epithelial transition (MET)-related genes such as *Ovol1* (Ovo Like Transcriptional Repressor 1), which is regulated in turn by the PI3K-AKT signaling pathway [34]. Jarid1b activates PI3K/AKT/Ovol1 by controlling suppressors of this pathway such as Ship1 (SH2-containing inositol-5′-phosphatase 1); knockdown of *Ship1* activates the downstream PI3K-AKT pathway and enhanced Ovol1 expression in HaCaT cells, whereas overexpression of Ship1 led to decreased Ovol1 expression [34]. Thus, Sun et al., demonstrated that Jarid1b negatively regulates Ship1 expression by directly marking its promoter to modulate H3K4me3 enrichment, and consequently, to drive the regulation of keratinocyte differentiation [34]. 

### 4.3. Epigenetic Regulation by the Genome Organizer Satb1

Special AT-rich sequence-binding protein-1 (Satb1) is a global chromatin organizer and transcription factor and is critical for integrating higher-order chromatin architecture with gene regulation [48]. 

Satb1 regulates gene expression by acting as a “docking site” for several chromatin remodelers and by recruiting corepressors or coactivators directly to promoter regions [48], thus establishing specific three-dimensional conformations in tissue-specific gene loci [49]. Satb1 is expressed in basal epidermal keratinocytes and promotes cell differentiation through higher-order chromatin folding and transcriptional regulation of the EDC locus [35], and its depletion in a mouse skin model caused thinning of the epidermis accompanied by strong alteration of the expression of terminal differentiation-associated genes in keratinocytes [35]. 

### 4.4. Epigenetic Regulation by DNA Methylation and Hidroxymethylation

DNMT1is both a maintaining and *de novo* DNA methylation enzyme that is enriched in undifferentiated epidermal progenitor cells, where it is required to retain proliferative potential and suppress differentiation; after the embryonic stages, it remains confined to the basal epidermal layer containing proliferating keratinocytes [50]. Khavari et al. used human 3D culture and DNMT1 knockdown in primary human keratinocytes to demonstrate a prominent role for DNMT1 in the maintenance of epidermal progenitor cells and epidermal tissue renewal [36]. 

Epidermal depletion of DNMT1 coexists with overexpression of genes associated with cell cycle arrest such as Cdk inhibitors (e.g., cycling-dependent kinase inhibitors p16INK4a and p15INK4B); however, cyclin-dependent kinase 4 and 6 (Cdk4/6) partially rescues the effect of DNMT1 impairment on cell proliferation [50]. 

These findings support the crucial role of DNMT1 in preserving epidermal progenitor cell identity and remodeling of this pattern during terminal differentiation [11]. 

Some gene promoters remain active when methylated through the binding of CCAAT-enhancer-binding proteins (C/EBPα) that contribute to the opening of the chromatin structure with consequent gene activation, whereas 5-aza-cytidine promotes the inhibition of DNA methylation and prevents the expression of a subset of genes during calcium-induced differentiation in culture [51]. 5-azacytidine also differentially affects gene expression within the EDC locus in normal human keratinocytes; indeed, Elder and Zhao demonstrated small proline-rich protein 1/2 (PRR1/2) and involucrin were overexpressed and S100A2 was under-expressed when compared to controls [52]. These data suggests that DNA methylation can regulate both gene activation and repression with a lack of correlation between methylation status and gene expression level, probably in line with the required balance between proliferation/quiescence of progenitor cells. 

Finally, 5-hydroxymethylcytosine (5hmC) is the first oxidative product in the active demethylation of 5-methylcytosine (5mC) (Figure 1a). Three ten-eleven translocation (TET) enzymes (TET1/2/3) catalyze the hydroxylation of DNA (5mC) to 5-hydroxymethylcytosine (5hmC) and can further catalyze oxidation of 5hmC to 5-formylcytosine (5fC) and then to 5-carboxycytosine (5caC) [53]. It has been proposed that 5-hydroxymethylcytosine, the first product in the active demethylation of 5mC, is prevalent in embryonic stems cells and reduced levels of TET1 and subsequently 5hmC causes the impaired self-renewal of stem cells [37]. It has also been hypothesized that conversion of 5mC to 5hmC by TETs blocks the repressive methyl-CpG-binding domain (MBD) and DNMT proteins [54], promoting the activation of gene expression; however, this role of this process in the control of gene expression in keratinocyte progenitor cells remains unclear. 

### 4.5. Epigenetic Regulation by Histone Deacetylation 

Histone deacetylases (HDACs) are a class of enzymes that remove acetyl groups from an ε-N-acetyl lysine amino acid on a histone, resulting in a more closed chromatin structure and repression of gene expression [55]. HDACs play a crucial role in coordinating the crosstalk between signaling pathways with chromatin remodeling and transcription factors to orchestrate gene expression [56]. HDAC1 and HDAC2 are involved in hair follicle formation as well as epidermal development and stratification. Their role in the control of the gene expression program in epidermal progenitor cells and epidermal differentiation has been demonstrated by LeBouef et al. who showed that HDAC1/2 null epidermis failed to differentiate, remaining single-layered [57]. The same authors also demonstrated that HDAC1/2 directly mediate the repressive activity of p63 in keratinocytes and independently suppressed p53 activity via deacetylation of the p53 protein [57]. 

Further evidence for the role of HDACs was obtained after depletion of HDAC1/2 from the basal layer of mouse epidermis resulted in increased histone acetylation and enhanced keratinocyte proliferation, leading to hyperplasia in addition to apoptosis and hair follicle dystrophy [38]. 

Zhu et al. demonstrated for adult mouse epidermis that homozygous epidermal HDAC1/2 codeletion—in association with a null p53 expression—partially restored epidermal proliferation, indicating that the effects of HDAC1/2 deletion on keratinocyte proliferation are mediated in part via increased p53 activity. Moreover, p16 deletion did not significantly rescue epidermal thickness, proliferation, and/or apoptosis of HDAC1/2 null epidermis, but its loss enabled survival of HDAC1/2-deficient keratinocytes by preventing their senescence. Finally, the same authors demonstrated that double *HDAC1/2* homozygous deletion caused hyperacetylation of c-Myc. Taken together, these findings indicate that HDAC1/2 play three distinct roles in adult epidermis: (i) deacetylation of p53 to promote progenitor cell proliferation, (ii) deacetylation of c-Myc to prevent premature differentiation, and (iii) restraint of p16 to repress senescence and allow long-term maintenance of progenitor cells [58]. 

### 4.6. Epigenetic Regulation by Polycomb Group (PcG) Proteins

Polycomb group (PcG) proteins are transcriptional repressors with a key role in stem cell identity and differentiation. PcG proteins are evolutionarily conserved and act within complexes, called Polycomb repressive complexes (PRCs).

PRC1 mono-ubiquitinates histone H2A on lysine 119 (H2AK119Ub1) and PRC2 catalyzes trimethylation of lysine 27 on histone H3 (H3K27me2/3) [59]. Chromobox 4 (Cbx4) is a component of the PRC1 complex and plays an important role in the maintenance of quiescence in human epidermal progenitor cells [60]. 

Bmi-1 is another PRC1 component. It is expressed in the basal layers of human epidermis and is implicated in the control of cell survival by altering cell cycle regulatory protein expression, such as p16 and p19, and inhibiting apoptosis [12,61]. 

PRC2 is the most widely conserved PcG complex in multi-cellular eukaryotes and is composed by four core subunits with an intrinsic histone methyltransferase activity; the catalytic subunit, known as Ezh2 (enhancer of zeste homolog 2), contains the signature SET domain commonly found in lysine methyltransferases [62]. It has been shown that Ezh2 as well as Ezh1 are involved in the regulation of EDC genes in epidermal cell progenitors, providing strong evidence for the involvement of this epigenetic actor in the temporal and spatial keratinocyte terminal differentiation program [63]. On the other hand, Ezh1/2 knockout mouse models do not show any epidermal defects, suggesting that the loss of their activities in epidermal keratinocytes might be compensated by other PRC2 components [63]. The transcriptional repressive function of Ezh1/2 are potentiated by Jarid2, which also acts as a component of the PRC2 complex [64]; depletion of Jarid2 in mouse epidermis inhibits proliferation and promotes differentiation of progenitor cells postnatally. Jarid2 deficiency in keratinocytes reduces H3K27m3, resulting in delayed hair follicle cycling because of decreased proliferation of hair follicle stem cells and their progeny [39]. Thus, similar to Ezh1/2, Jarid2 is required for the maintenance of cell proliferation and inhibition of differentiation in epidermal stem and progenitor cells [39]. 

### 4.7. Epigenetic Regulation by microRNAs

MicroRNAs (miRNAs) are a group of short (~22 nucleotides), single stranded, non-coding RNAs, which can regulate the expression of other protein-coding gene at post-transcriptional level. In most cases, miRNAs interact with the 3′ UTR of target mRNAs to suppress expression [65], but they can also activate gene expression under specific conditions [25]. In human skin, different miRNAs have been reported to play a pivotal role in epidermal development by regulating differentiation [66], and several studies have provided evidence of differential patterns of miRNAs expression in epidermis and hair follicle, which may be indicative of differences in the epigenetic control of these two populations [67]. 

There are several reports documenting the role of miRNAs in epidermal development and differentiation; for example, miR203, is induced in terminally differentiated keratinocytes, both in vitro and in vivo. Through the regulation of p63 expression, miR203 maintained the proliferative and differentiation potential of basal keratinocytes as well as precursor cells [68]. Besides miR-203, miR-23b was found to be a second miRNA highly expressed in differentiated keratinocytes, identifying it as a differentiation marker for human skin [69].

A total of nine miRNAs—miR-203, miR-95, miR-210, miR-224, miR-26a, miR-200a, miR-27b, miR-328, and miR-376a—were found not only to be induced in terminally differentiated keratinocytes in vitro and in vivo but were also functionally involved in the differentiation process [69]. 

Another miRNA, miR-214, when overexpressed, led to a reduction in epidermal thickness, lower keratinocyte proliferation rate, and hair follicle loss [69]. 

These examples suggest that several miRNAs are involved in different processes between proliferation and differentiation in the program of epidermal development. 

## 5. The Role of Epigenetics in Inflammatory Skin Diseases

In the field of dermatology, skin disorders notably affect skin integrity and impair epidermal differentiation. Epigenetic mechanisms have been extensively studied in cutaneous T-cell lymphoma (CTCL) [70], but in recent years, the causative role of epigenetics is also emerging in inflammatory skin diseases, such as psoriasis [71] and atopic dermatitis [72]. The involvement of epigenetic determinants in these conditions has been arduously studied, and the great amount of published data reflects their importance in the etiology of these diseases. A dysregulation of multiple epigenetic mechanisms, including aberrant DNA methylation, alterations in histone modifications, and miRNA expression have been found to play a crucial role in the pathogenetic scenarios of these skin conditions. Notably, differences in epigenetic signatures could be observed not only between the skin of healthy control and lesional skin of psoriatic patients but also between a patient’s lesional, perilesional, and unaffected skin [73,74,75,76], indicating different phases of the disease may have unique epigenetic signatures.

### 5.1. The Role of Epigenetics in Psoriasis 

Psoriasis is a chronic, immune-mediated inflammatory skin disease with a multifactorial etiology involving accelerated proliferation and abnormal differentiation of keratinocytes [77]. It is now considered an autoinflammatory keratinization disease (AIKD), a term that encompasses disorders with mixed pathomechanisms of autoinflammation and autoimmunity [78]. 

Epigenetic changes, including DNA methylation, histone modification, and noncoding RNA regulation have been reported to be involved in the complex pathogenesis of psoriasis (Figure 2). 

#### 5.1.1. DNA Methylation 

Roberson et al. first identified global DNA methylation in psoriatic skin lesions when compared to controls, in which more than 1000 differentially methylated CpG islands (CGIs) were detected and, among them, twelve mapped to the epidermal differentiation complex or nearby genes upregulated in psoriasis. Moreover, the authors highlighted the reversible nature of DNA methylation, suggesting that this epigenetic mechanism is a dynamic actor in this disease [79]. 

Chandra et al., in an epigenome-wide DNA methylation study, found several differentially methylated loci, overlapped with different PSORS regions, involved in the regulation of pathogenetic gene expression, such as *S100A9* (S100 calcium binding protein A9), *SELENBP1* (selenium binding protein 1), *CARD14* (caspase recruitment domain family member 14), *KAZN* (kazrin, periplakin interacting protein), and *PTPN22* (protein tyrosine phosphatase non-receptor type 22), indicating the potential role of DNA methylation in regulating specific histopathological features commonly seen in psoriasis [80]. 

A candidate gene approach has been employed to define the promoter methylation profile of several psoriasis risk loci and, according to gene ontology, CpG hyper/hypomethylation coincided with genes involved in many processes impaired in the pathogenesis of psoriasis, such as cell cycle, apoptosis, immune system regulation, cell communication, and signal transduction [81]. Loss of methylation has been unraveled at the promoter region of *SHP-1* (Src homology region 2 domain-containing phosphatase-1), a protein tyrosine phosphatase (PTPs) that regulates several cellular processes, including growth and cell differentiation [82], and at the promoter level of p16 [83], p15 and p21, which encode negative regulators of the cell cycle in hematopoietic stem cells of psoriatic patients [84]. 

In addition, hypermethylation has been detected in the promoter sequences of *PDCD5* (programed cell death 5) and *TIMP2* (TIMP metallopeptidase inhibitor 2) genes, which are involved in apoptosis and the maintenance of tissue homeostasis by suppressing the proliferation of quiescent tissues, respectively, thus regulating the proliferation of keratinocytes [81]. Moreover, the promoter of *SFRP4* (secreted frizzled related protein) gene, a negative regulator of the Wnt signaling pathway, has been found heavily hypermethylated in psoriasis skin lesions and its consequent downregulation contributes to epidermal hyperplasia [85].

Finally, genome-wide association studies have identified hundreds of hypermethylated genes in psoriasis, and among them are immune-associated genes, such as *TLR-7* (Toll-like receptor 7) and *IRAK1* (interleukin 1 receptor associated kinase 1) [86].

#### 5.1.2. Histone Modification 

Evidence of unbalanced histone modification in psoriatic skin is also beginning to increase. H3K9 dimethylation is decreased in keratinocytes from psoriasis patients, correlating with IL-23 overexpression and supporting its relevance for this disease. Indeed, the actin polymerizing molecule N-WASP controls IL-23 expression in keratinocytes in response to TNF-α by regulating the degradation of the histone methyltransferases G9a and GLP and H3K9 dimethylation of the IL-23 promoter, leading to a chronic skin inflammation with an IL-23 inflammatory profile [76]. Zhang et al. demonstrated that mRNA expression levels for histone deacetylase HDAC1, histone methyltransferase SUV39H1, and EZH2 were all increased in psoriatic peripheral blood mononuclear cells (PBMCs), correlating with higher keratinocyte proliferation, and contributing to psoriatic hyperplasia [87]. It has also been reported that H3K27 demethylation, via Jmjd3, regulates Th17 cell differentiation and expression of several inflammatory cytokines, establishing a pathogenetic role for this epigenetic mechanism in psoriasis; indeed, Jmjd3 directly bound to and reduced H3K27 trimethylation levels in the genomic region of *Rorc* (RAR related orphan receptor C), which encodes the master Th17 transcription factor Rorγt and Th17 cytokine genes such as IL-17, IL-17f, and IL-22 [88].

#### 5.1.3. MicroRNAs 

Changes in miRNA expression have also been found in psoriatic skin lesions and more than 250 miRNAs have been implicated in the pathogenesis of this skin disorder [89]. MiR-203 is exclusively expressed by keratinocytes and its upregulation in psoriatic skin lesions is associated with the downregulation of its target, *SOCS-3* (suppressor of cytokine signaling 3), which is involved in inflammatory responses and keratinocyte functions [89]. Subsequently, Zibert et al. discovered one downregulated and nine upregulated miRNAs in non-lesional psoriatic skin, including miR-21, miR-205, miR-221, and miR-222, suggesting their role in early phases of psoriasis [90]. Moreover, inhibition of miR-21 reduces disease severity in patient-derived psoriatic skin xenotransplants in mice as well as in a psoriasis-like mouse model, suggesting that miR-21 could represent a potential therapeutic target for the treatment of psoriasis [91]. 

MiR-146a is another microRNA that appears upregulated in psoriasis. It can promote TNF expression and is positively correlated with IL-17-driven inflammation in keratinocytes, revealing a key role with therapeutic potential in the pathogenesis of psoriasis [92,93]. In addition, miR-155 has been identified as a potential therapeutic target for psoriasis [94]; indeed, it is markedly increased in lesional skin and PBMCs of psoriasis patients and plays several crucial roles in keratinocyte proliferation and apoptosis inhibition—through the phosphatase and tension homolog deleted on chromosome 10 (PTEN) signaling pathway—as well as in inflammatory pathways [94,95,96]. MiR-31 and miR-210 have also been shown to contribute to inflammation in psoriatic skin plaques by regulating the production of inflammatory cytokines/chemokines, enhancing keratinocyte proliferation [97], and inducing Th17 and Th1 cell differentiation [98], respectively. 

Some miRNA clusters have also been also implicated in the pathogenesis of psoriasis; for example, cytokine-induced overexpression of the miR-17-92 cluster can promote keratinocyte proliferation, contributing to the development of psoriasis-like inflammation [98]. 

Finally, serum levels of miR-33, miR-126, and miR-143 have been proposed as potential biomarkers for disease severity or the prognosis of psoriasis [99]. 

### 5.2. The Role of Epigenetics in Atopic Dermatitis

Atopic dermatitis is one of the most common inflammatory skin diseases worldwide with a recurrent, chronic-relapsing clinical course characterized by an impairment of the skin barrier function supported by the (epi-)dermal immune system and the microbiome of the skin [100]. 

Epigenetic modifications have also been recognized in this disease and are mainly mediated by DNA methylation and non-coding RNAs [101,102] (Figure 3). 

#### 5.2.1. DNA Methylation

Regarding DNA methylation changes in atopic dermatitis, Rodriguez et al. [103] used an epigenome-wide approach to identify 127 differentially methylated CpG sites (DMSs) between atopic dermatitis lesional skin and controls. Methylation in several of these regions correlated with genes encoding keratins located within the keratin cluster and altered S100A gene expression within the EDC [103]. The same authors confirmed S100A2, A7, A8, A9, and A15 overexpression in atopic dermatitis lesional skin; these expression changes correlated with DNA hypermethylation of one CpG within S100A5, indicating a coregulatory mechanism through methylation. Similarly, KRT6A and KRT6B mRNA overexpression depended on decreased methylation of a single CpG in *KRT6A* [103]. Both these keratins belong to the group of “alternative pathway keratins”, and it is supposed that this alternative pathway provides a physiological response for epidermal wound healing after injury [104]. A single differentially methylated CpG site within OAS2 (2′-5′-oligoadenylate synthetase 2) coupled with altered co-expression was also observed for OAS1 (2′-5′-oligoadenylate synthetase 1), OAS2, and OAS3 (2′-5′-oligoadenylate synthetase 3) [104], which belong to a family of proteins—induced by interferons—that synthesizes 2′,5′-oligoadenylates and are involved in the innate immune response to viral infection [105]. Further, CD36 has been found upregulated upon skin barrier disruption [106] as well as in other inflammatory skin conditions [107,108] that contribute to terminal keratinocyte differentiation [107]. 

Collectively, these findings have confirmed—in lesional skin of atopic dermatitis patients—an altered methylation pattern affecting key genes for keratinocyte differentiation, proliferation, and the immune response [104].

The *TSLP* (thymic stromal lymphopoietin) gene encodes a hemopoietic cytokine that plays a crucial role in promoting a T helper type 2 (TH2) cell response. It appears to be a central player in the development of asthma and atopic dermatitis and is being considered as a potential therapeutic target for the treatment of such diseases [109]. In a study conducted by Luo et al. [110], the authors investigated *TSLP* methylation status, and the results showed—in lesional skin from patients with atopic dermatitis when compared to control—hypomethylation of the *TSLP* gene at the promoter level, confirming that DNA hypomethylation contributes to TSLP overexpression. 

In another study conducted by Olisova et al., DMSs were observed within genes involved in several atopic dermatitis-related processes, including immune response regulation, lymphocyte activation, cell proliferation, apoptosis, and epidermis differentiation, highlighting marked epigenetic involvement in the development of this disease [111]. 

Finally, a study investigating the relationship between genetic and epigenetic changes in the filaggrin (*FLG*) gene—essential for the regulation of epidermal homeostasis—in PBMCs of patients with atopic dermatitis, indicated that the association between loss-of-function mutations in the *FLG* gene and eczema was orchestrated by DNA methylation; in particular, at an 86% methylation level, filaggrin haploinsufficient subjects had about a 6-fold increased risk of eczema when compared to those with wild type *FLG* [112]. 

#### 5.2.2. MicroRNAs

Several microRNAs have an active part in the pathogenetic scenario of atopic dermatitis. Upregulation of miR-10a-5p has been demonstrated to impair keratinocyte proliferation and migration through HAS3 (hyaluronan synthase 3), a damage-associated positive regulator of keratinocyte proliferation and migration that is the direct target of miR-10a-5p [113]. Gu et al. demonstrated miR-29b upregulation in lesional skin and sera from atopic dermatitis patients. This microRNA promotes keratinocyte apoptosis by inhibiting Bcl2L2 (Bcl-2-like protein 2) protein, contributing to epithelial barrier dysfunction typically impaired in atopic dermatitis [114]. 

Other microRNAs that play a critical role in atopic dermatitis are the following: (i) miR-124 that regulates inflammatory responses in keratinocytes and chronic skin inflammation through the NF-κB pathway [115]; (ii) miR-143 that decreases IL-13 activity and the inflammatory cascade through targeting IL-13Rα1 in epidermal keratinocytes [116]; (iii) miR-146, a mediator of inflammation, which is regulated by inflammatory mediators such as interleukin 1 and TNF-α and operates in a “negative regulatory loop” to modulate the inflammatory response [117]; (iv) miR-155, which regulates both cytokine responses and epithelial barrier function by targeting PKIα [118]. Among these, miR-124 and miR-143 can be regarded as potential novel therapeutic targets in AD patients [115,116]. 

Interestingly, miR-17-5p, miR-20a, miR-21, and miR-106b are upregulated in both atopic dermatitis and psoriasis skin lesions, whereas miR-122a, miR-133a-133b, miR-133b, miR-215, and miR-326 are downregulated in both diseases [89]. The similar expression of miRNAs in both atopic dermatitis and psoriasis lesions is consistent with the common clinical features of these diseases (Figure 4).

## 6. Conclusions 

The central dogma of biology holds that cell information flows from DNA to RNA to proteins [119]; this concept has now been completely confuted due to the role of epigenetics in regulating gene expression. The Greek prefix epi- (ἐπι- “over, outside of, around”) in epigenetics implies features that are “on top of” or “in addition to” the traditional genetic basis for inheritance [120], and the term “epigenetic” has become one of the newest emerging and interesting fields in the scientific world. Epigenetics refers to enzymatic modifications—DNA methylation, histone acetylation/deacetylation, and RNA regulation—to the chromatin structure that can up- or down-regulate gene expression in the absence of DNA sequence changes [18]. More and more data demonstrate that epigenetic mechanisms are involved in the regulation of multiple aspects of epidermal growth and differentiation. Moreover, epigenetic modifications can be activated by environmental factors, upon which their action appears to help epidermal cells adapt to long-term physiological changes [121]. To note, the skin neuroendocrine system acts by defending and maintaining the structural and functional integrity of the skin as well as systemic homeostasis [122]. Epigenetic regulators modulate both local and higher-order chromatin structure in epidermal keratinocytes, affecting the function of the genes, mainly those in EDC, that are associated with proliferation and differentiation processes in keratinocytes as well as their progenitor cells.

DNA methyltransferase DNMT1, histone deacetylases HDAC1/2, and Polycomb components (Cbx4, Bmi1, Ezh1/2) act as repressive chromatin regulators stimulating proliferation of progenitor cells, and some of these also inhibit premature activation of terminal differentiation-associated genes. In contrast, a group of chromatin remodelers such as histone demethylases, ATP-dependent chromatin remodeler Brg1, and genome organizer Satb1 promote terminal keratinocyte differentiation and also exhibit effects on cell proliferation. Finally, microRNAs are also involved in different processes between proliferation and differentiation in the program of epidermal development. 

As demonstrated by several studies concerning skin diseases, changes in the epigenetic signature in the epidermis can contribute to the pathogenesis of these diseases. More importantly, in contrast to genetic changes—which are difficult to reverse—epigenetic modifications are pharmaceutically reversible; thus, the emerging tools of epigenetics can be used as preventive, diagnostic, and therapeutic markers [123]. 

In the past few decades, development of epigenetic drugs—so-called “epidrugs”—has achieved significant progress; thus, epigenetic therapy is a promising new therapeutical solution that targets the main causative epigenetic mechanisms involved in inflammatory/autoinflammatory diseases. Future efforts in this direction will help to bridge the gap between our current knowledge and potential applications for epigenetic regulators in clinical practice to pave the way for promising tailored treatments.

## Figures and Tables

**Figure 1 ijms-23-04874-f001:**
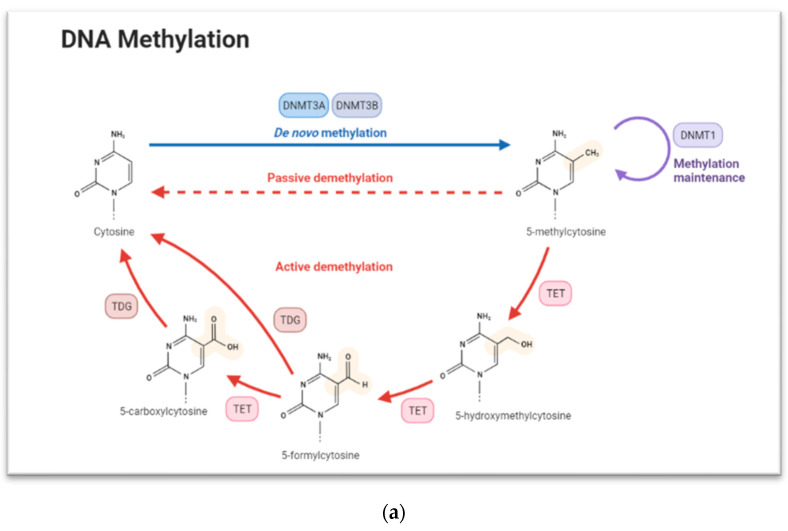
(**a**) The concurrence of DNA methylation and demethylation. Exported by Biorender.com (accessed on 1 April 2022). (**b**) Histone modifications including acetylation and methylation of lysine. Created in Biorender.com. (**c**) MicroRNA formation and their role in protein degradation. Created in Biorender.com.

**Figure 2 ijms-23-04874-f002:**
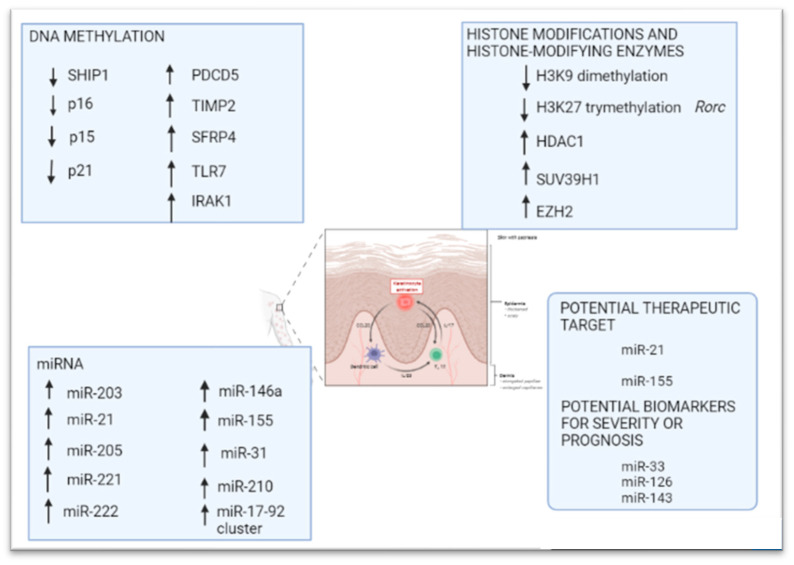
Epigenetic modifications in psoriasis: DNA methylation, histone modifications and microRNAs, including the microRNAs that can be considered as potential therapeutic targets and biomarkers for severity or prognosis. Created in Biorender.com.

**Figure 3 ijms-23-04874-f003:**
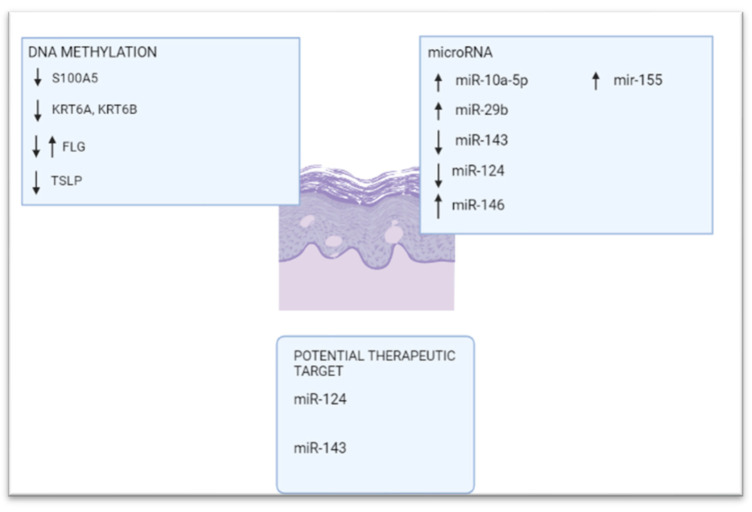
Epigenetic modifications in atopic dermatitis: DNA methylation and microRNAs, including the microRNAs that can be considered as potential therapeutic targets. Created in Biorender.com.

**Figure 4 ijms-23-04874-f004:**
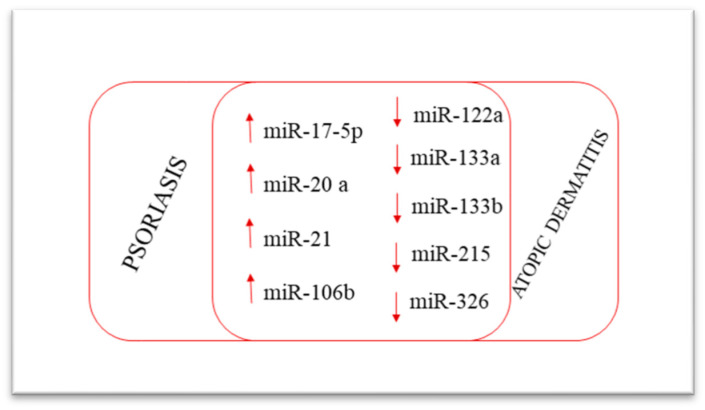
MicroRNAs involved in both psoriasis and atopic dermatitis.

**Table 1 ijms-23-04874-t001:** Effect of knockout/down and overexpression of chromatin-modifying protein on keratinocyte/epidermal growth and differentiation.

Gene/Modified Gene	Effect on Chromatin	Effect on Keratinocyte/Epidermal Growth and Differentiation
Brg1	Alteration of the chromatin architecture by repositioning, assembling, and restructuring nucleosomes	Overexpression of the EDC genes [26]
Brg1 knockout	Alteration of the chromatin architecture by repositioning, assembling, and restructuring nucleosomes	Impairment of the final stage of keratinocyte terminal differentiation [27]
SWI/SNF complex	Alteration of the chromatin architecture by repositioning, assembling, and restructuring nucleosomes	Suppress differentiation and promote the maintenance of the progenitor state in epidermal cells [28]
Mi-2β	Alteration of the chromatin architecture by repositioning, assembling, and restructuring nucleosomes	Self-renewal of epidermal precursors during the earlier phases of embryogenesis; defective basal layer formation [29]
Mi-2β knockout	Alteration of the chromatin architecture by repositioning, assembling, and restructuring nucleosomes	Impaired induction and development of the hair follicles [29]
JMJD3 knockdown	H3K27me3 ↑	Blocked progenitor cell differentiation [30]
JMJD3 overexpression	H3K27me3 ↓	Enhanced expression of epidermal differentiation markers [30]
Setd8 knockout	H3K20me1 ↓	Inhibition of progenitor cell proliferation; impaired differentiation [31,32]
Suv39H1 knockout	H3K9me3 ↓	Induction of genes encoding differentiation markers [33]
Jarid1b knockdown	H3K4me3 ↑	Delayed differentiation [34]
Jarid1b overexpression	H3K4me3 ↓	Reduced proliferation; enhanced differentiation [34]
Satb1	Integration of higher-order chromatin architecture with gene regulation	Induction of cell differentiation [35]
Satb1 knockout	Integration of higher-order chromatin architecture with gene regulation	Altered expression of terminal differentiation-associated genes [35]
DNMT1 knockdown	DNA methylation	Maintenance of epidermal progenitor cells and epidermal tissue renewal [36]
5-hydroxymethylcytosine	Demethylation 5mC	Impairment of self-renewal of stem cells [37]
HDAC1/2 knockout	acH3 ↑	Enhanced proliferation; epidermal hyperplasia; disturbed hair follicle differentiation [38]
Jarid2 knockout	H3K27me3 ↓	Inhibition of proliferation; premature differentiation [39]

## Data Availability

Not applicable.

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
