# Peer review of "Epigenetic Mechanisms of Epidermal Differentiation"

_ijms, 2022, doi:10.3390/ijms23094874_

Round 1
Reviewer 1 Report
Title: " Epigenetic mechanisms of keratinocytes proliferation and differentiation”
Authors: Chiara Moltrasio, Maurizio Romagnuolo, Angelo Valerio Marzano
Comments:
The purpose of this review is to summarize the available evidence on activating and repressive effects on chromatin that promote keratinocyte differentiation and progenitor cell proliferation. The review does not read smoothly, as for the most part only references were strung together and it contains many abbreviations. Better transitions and more of the reader's own thoughts should be incorporated, e.g. a self-formulated introductory and concluding sentence to each paragraph. Also, the chapters should be structured in such a way that the reader first gets an overview of the topic and then the details are explained.
Major points:
- Title is a bit unspecific and meaningless.
- Page 1 line 38-40: Introduction is not scientific, you read about molecular details before the topic of the paper is clear. Introduction should be more general.
- Page 1 line 43: Here I miss a reference that proves the statement also for "skin disorders".
- Introduction general: it is not really described for what the topic of the title and thus of the review is important and for what the summary (see page 2 line 63-65) can be helpful.
- Page 2, lines 76/77: The introduction to the topic is too abrupt. I suggest to insert a general heading before 3.1. (e.g. The role of epigenetics in healthy tissue) and a short paragraph with the information that physiological processes will be described in the following.
- Page 3, line 137: Please add information about which cells HaCaT is about.
- The outline with "2. methods", "3. results" etc. does not read very nicely as an overview. Titles for the chapters that reflect the content would be more appropriate and interesting; also there is no transition between the chapters, it seems more like a list (runs through the whole work).
- In the chapter "3. Results" it starts again directly with details; I miss an introduction and embedding in the actual topic: a connection is not clear.
- The structure is not quite logical: I would start chapter 3.8. "The role of epigenetics in inflammatory skin diseases" before the individual mechanisms are described in detail. I would first explain the general relationships, then the mechanisms, and then give examples based on the diseases in question.
- In chapter 3.8. the already mentioned mechanisms (page 6 line 276) like "micro RNAs" are repeated again as subchapters (and even 2x!!: line 364 and line 410). Are the pathways summarized here currently used therapeutically and if so, how? Please add information.
- Page 7, line 302: Please add what the main findings are in this area.
- Page 7 line 312; page 8 line 387: "DNA" must be capitalized
- Why are only psoriasis and atopic dermatitis treated? What about other skin diseases?
- A table summarizing the different epigenetic mechanisms would be helpful.
- Page 9, line 400, correction of reference: it should be Olisava instead of Olislova.
- There are no figures or tables at all; that could be done well with the mechanisms. A figure showing which epigenetic signaling pathways occur in healthy individuals and what is disturbed in the diseases mentioned.
- There is no list of abbreviations.
- Conclusion should be substantially revised: I miss a meaningful conclusion + discussion on the topic (again, it tends to go into details).
Reviewer 2 Report
The topic is of interest. However the paper requires revisions in the content and reference list. Specifically, 89 citations for an expert review on the topic is unacceptable. This number should triplicate.
The subject is very narrow and should be extended to local regulatory molecules regulating differentiation program. For example that there is a coordinated local regulatory activity to counteract environmental stressors insults through differentiation program (Adv Anat Embryol Cell Biol. 2012;212:1‐115. PMCID: PMC3938165).
I am surprised that the authors ignore diverse homeostatic actions of the UVR (Endocrinology 159(5), 1992-2007, 2018).
Also reviewers would expect a schematic figure outlining the concept.
In summary, the choice of regulatory subjects is too narrow and low number of reference for an expert review is unacceptable.
Round 2
Reviewer 1 Report
The authors have considered only a part of the comments.
I still miss in this work the precise and broad range of regulatory mechanisms, which is not sufficiently justified scientifically.
I miss in the whole paper the images 1.1, 1.2, 1.3, 2 and 3. In addition, I miss the table 1.
Reviewer 2 Report
The authors adequately revised the manuscript and replied to the critique.
They may consider mentioning of the neuroendocrine activities of the skin which would be of interest to the readers (Endocrine Rev 21, 457-487, 2000)
Round 3
Reviewer 1 Report
The authors have satisfactorily addressed the concerns raised in the original version. The revised version is significantly improved. No further concerns.